# Investigation of the Effects of Metallic Nanoparticles on Fertility Outcomes and Endocrine Modification of the Hypothalamic-Pituitary-Gonadal Axis

**DOI:** 10.3390/ijms241411687

**Published:** 2023-07-20

**Authors:** Miguel A. Sogorb, Héctor Candela, Jorge Estévez, Eugenio Vilanova

**Affiliations:** Instituto de Bioingeniería, Universidad Miguel Hernández de Elche, Avenida de la Universidad s/n, 03202 Elche, Spain; msogorb@umh.es (M.A.S.); hcandela@umh.es (H.C.); jorge.estevez@umh.es (J.E.)

**Keywords:** nanoparticle, nanotoxicology, fertility impairment, gonadotropin-releasing hormone, hypothalamic-pituitary-gonadal axis, silver nanoparticle, titanium dioxide nanoparticle, quantum dot nanoparticle

## Abstract

Nanotechnology is a very disruptive twenty-first-century revolution that will allow social and economic welfare to increase although it also involves a significant human exposure to nanoparticles. The aim of the present study was to contribute to the elucidation on whether metallic nanoparticles have a potential to induce fertility impairments. Regulatory studies that observed official OECD guidelines 415, 416 and 422 have failed to detect any fertility alterations caused by nanoparticle exposure. However, the scientific literature provides evidence that some nanoparticles may cause gonad impairments although the actual impact on fertility remains uncertain. This aim of the present study is to revisit the previously published RNAseq studies by analyzing the effects of several nanoparticles on the transcriptome of T98G human glioblastoma cells given that glial cells are known to play a pivotal role in the regulation of gonadotropin releasing hormone neurons. We found evidence that nanoparticles impair the gonadotropin releasing hormone receptor pathway and several related biological process like, among others, the cellular response to follicular stimulating hormone, cellular response to gonadotropin stimulus, cellular response to hormone stimulus, response to steroid hormone, ovulation cycle and response to estradiol. We propose that nanoparticles interfere with the ability of glial cells to regulate gonadotropin-releasing hormone neurons and, subsequently, the hypothalamic-pituitary-gonadal axis, potentially leading to fertility impairments. To our knowledge, this is the first proposal of a mode of action based on endocrine disruption for explaining the possible effects of nanoparticles on fertility. Whether these finding can be extended to other types of nanoparticles requires further investigation.

## 1. Introduction

Nanotechnology promises scientific advances in many fields. Nanotechnology is based on nanoparticles (NPs), which are defined as those particles smaller than 100 nm in at least one dimension. The nanosize of the particles confers “smarter” physical-chemical properties (higher electrical or thermal conductivity, flexibility, resistance and the ability to cross biological membranes) than equivalent materials on a non-nanometric scale. Thus, nanotechnology applications have been described in the following fields, among others: food industry (food packaging and additives); biomedicine (site-directed drug delivery, diagnosis through imaging techniques and nanoneedles); environment (water treatment and energy storage); pigments (paints, printer inks and tattoos); cosmetics (UV filters). Nanotechnology has even been proposed to combat the COVID-19 pandemic with applications, such as disinfectants, personal protective equipment, diagnostic systems and vaccine development [1,2]. According to the European Union Observatory for Nanomaterials (EUON) on 18 May 2023 it was determined that there were 343 different nanomaterials on the market of the European Union [3].

Overall, all the above-stated data and information allow us to predict marked human exposure to NPs. However, the safety of an application should be warranted before a use is approved, including occupational, environmental and consumer uses. Indeed, there have been proposals to harmonize the testing of nanomaterials for European Union regulatory requirements on chemical safety, which note that further work on OECD Technical Guidelines (OECD-TG) 415, 416, 421, 422 and 443 is needed (to specifically address both the dispersion of NPs and the measurement of NPs in biological tissues) [4]. It is worth noting that new approaches based on cellular test methods may also be relevant to support hazard identification of NPs and to reduce animal testing specifically for fertility and reproductive toxicity, as these OECD TGs require the largest number of animals of those used to test for human hazards. Larsen and co-workers [5] reviewed the studies on reproductive toxicity of nanomaterials in response to a request from EUON. They concluded that there are conflicting data on the effects on male fertility after exposure to NPs. The studies reported some type of noted effect on sperm impairment (production, morphology and motility). However, these effects were reported in non-guideline studies for most of the nanomaterials investigated, and the authors did not rule out the possibility that, in certain cases, effects could be explained by the chemical composition of the NPs rather than their nanosize. Larsen and co-workers [5] also highlighted that the available OECD-TG studies did not report any adverse effects on male reproductive performance or fertility parameters. Studies published after this meta-analysis point in the same direction, as is the case on impairments of the mitochondrial dynamics in mouse testis after five intravenous doses (spaced with 3 day intervals) of 5 mg/kg body weight of TiO_2_- and Ag-NPs [6]. In an extended one-generation reproductive toxicity study (which was not conducted according to OECD-TG guidelines), Hong and co-workers [7] reported that 350 mg/kg body weight of ZnO-NPs caused statistically significant reductions in conception and pregnancy rates, sperm survival and sperm count, presumably by disrupting the blood-testis barrier and destroying the microenvironment needed for spermatogenesis. Silver NPs were found able to interfere with spermatogenesis and oogenesis by negatively affecting meiosis in zebrafish gonads [8] and ZnO-NPs were also able to cause follicular developmental retardation, which alters oocyte ovulation and “reducing the fecundity of female zebrafish [9].

Hypothalamic-pituitary-gonadal axis (HPGA) disruption leads to a type of infertility called hypogonadotropic hypogonadism, characterized by the impairment of reproductive maturation and function [10]. Under normal HPGA function (Figure 1), hypothalamus neurons release gonadotropin-releasing hormone (GnRH) into the portal pituitary veins. GnRH acts on the gonadotroph cells of the pituitary and stimulates the release of luteinizing hormone (LH) and follicle-stimulating hormone (FSH). LH and FSH act on testes and ovaries by bringing about the release of testosterone, progesterone and estradiol. These latter three sex hormones induce negative feedback on the HPGA, inhibiting the release of GnRH, LH and FSH [10].

The absorption of NPs into the central nervous system is not efficient due to the tightly packed cells forming the blood-brain barrier. Therefore, translocation of NPs into the central nervous system through the olfactory bulb appears to be the most effective route [11]. It is not clear how NPs can exit the brain, but it is conceivable that cerebrospinal fluid may act as a vehicle to release NPs into the circulating blood, and subsequently, the NPs would be excreted from the body in urine (for sizes smaller than 5 nm) and feces [11].

Glia are non-electrical cells found in the nervous system and consist of several cell types, such as astrocytes (involved in modulating synaptic transmission), microglia (specialized macrophages that protect central nervous system neurons) and oligodendrocytes (involved in myelination) [12]. Glia represent approximately 80% of the cells in the human brain and are involved in the homeostasis of the vast majority of neurobiological processes [13].

Glial cells play a pivotal role in the GnRH neuron development during embryonic and perinatal life and, although mechanisms are still unknown, are also involved in hypogonadotropic hypogonadism and other syndromes causing fertility impairments [14]. It is also known that glial cells, in particular astrocytes and tanycytes, regulate neuroendocrine functions (Figure 1) with a marked role in morphological plasticity, production of gliotransmitters or metabolic pathways that directly influence the actions of the neurohormones [14]. The role of glial cells is especially active in the control of GnRH, that is regulated via the gliotransmitters prostaglandin E2 [15]. Another experimental evidence, which plays in favor of the hypothesis that glia regulate GnRH neurons is the fact that selective depletion of the glial fibrillary acidic protein produced by tanycytes in transgenic mice caused a marked decrease in the testosterone levels and testicular weight, hypogonadotropic hypogonadism condition and a sharp decrease in the number of GnRH neurons [16]. Taking these findings into consideration it is suggested that a multiformal human glioblastoma as T98G cells could be a suitable in vitro model for studying the perturbations in the GnRH neuron performance derived from exposure to NPs and other xenobiotics.

We previously characterized the effects of ZnO- and TiO_2_-NPs (both with very similar applications, including pigments, cosmetics, imaging diagnostics, drug delivery vehicles, or dental composites) [17]; Ag-NPs (used in textile materials to enhance antimicrobial finishes, as an antimicrobial, drug delivery carrier, wound and bone healing or cosmetic) [18] and Cd/Se quantum dots (Cd/Se-QDs) NPs (used in imaging, sensing and drug delivery) [19] on T98G human glioblastoma cells through massive parallel RNA sequencing (RNAseq). We found alterations in many of the molecular pathways in all cases, except for the ZnO-NPs. We reported notable alterations in processes of inflammation, blood brain barrier integrity and hypothalamus regulation. We specifically found alterations in the GnRH receptor pathway for Cd/Se-QDs- and Ag- and TiO_2_-NPs, although we did not conduct a detailed analysis of these alterations at that time. After observing similar alteration by several NPs, we reassess here the information and note that these alterations in the GnRH receptor pathway and other biological processes related to the HPGA are commonly observed in several NPs. Thus, they could indeed be relevant for the fertility hazard identification of NPs and its mode of action. Overall, the aim of the present study is to provide knowledge to elucidate on whether metallic NPs can induce fertility disorders, altering male or female reproductive functions in aspects, such as conception, sperm quality, gonadal impairments, alterations in female reproductive cycle and other usual evaluations, for example, in OECD TG 421. In order to reach this goal, we reassess our previous RNAseq data by focusing on the possibility that some NPs could induce alterations, which could be related with possible fertility impairments via an endocrine disruption mode of action based on the alteration of the HPGA.

## 2. Results

### 2.1. The GnRH Receptor Pathway

According to the Protein ANalysis THrough Evolutionary Relationships (PANTHER) knowledgebase [20], the GnRH receptor pathway in *Homo sapiens* comprises 237 genes (listed in Appendix A). The Kyoto Encyclopedia of Genes and Genomes (KEGG), a database for taxonomy-based analysis of pathways and genomes [21], describes how a GnRH receptor is coupled to G-proteins and other secondary messengers, such as diacylglycerol, inositol 1,4,5-trisphosphate and protein kinase C, which provokes the release of intracellular calcium (Figure 2). Downstream of protein kinase C, the epidermal growth factor (EGF) receptor and mitogen-activated protein kinases (MAPKs), including extracellular-signal-regulated kinase (ERK), Jun N-terminal kinase and p38 MAPK, are also activated (Figure 2, see full picture for the whole route in Appendix A) [22].

### 2.2. Effectof ZnO-NPs on Hypothalamic-Pituitary-Gonadal Axis in T98G Human Glioblastoma Cells

Fuster and co-workers [17] exposed T98G human glioblastoma cells for 48 h to 5 µg/mL of ZnO-NPs. This concentration was the highest resulting in a reduction in a cell viability of 10%. The authors reported no differentially expressed genes (DEGs) after exposure. Therefore, under these experimental conditions, ZnO-NPs did not cause alterations to the HPGA.

### 2.3. Effect of TiO_2_-NPs on Hypothalamic-Pituitary-Gonadal Axis in T98G Human Glioblastoma Cells Exposed to

Fuster and co-workers [17] exposed T98G human glioblastoma cells for 48 h to 20 µg/mL of TiO_2_-NPs. This non-cytotoxic treatment dysregulated the expression of 1025 genes: 126 of them had fold-changes (FC) of less than 0.5, 82 had an FC greater than 2 and 817 had intermediate fold-changes of between 0.5 and 2. We uploaded the complete list of DEGs (taken from [17]) to the Gene Ontology (GO) resource and performed statistical overrepresentation tests using PANTHER pathways and GO biological processes as annotations datasets. The results are summarized in Table 1 and Table 2.

The genes belonging to the GnRH receptor pathway showed a 2.5-fold overrepresentation in the set of DEGs after exposure to TiO_2_-NPs (Table 1). No other PANTHER pathways, except cholesterol biosynthesis, were detected with statistical significance. However, it was noted that the genes belonging to the transcription growth factor (TGF) β signaling pathway were overrepresented (enrichment factor (EF) = 2.6) with a statistical *p*-value of 2.4 × 10^−3^, which corresponded to a False Discovery Rate (FDR), slightly higher than the threshold of 0.05 (7.8 × 10^−2^).

Ten different GO biological processes, potentially involved in the regulation of the HPGA, were overrepresented among the set of DEGs after exposure to TiO_2_-NPs. These biological processes were “*cellular response to FSH stimulus*”, “*negative regulation of ERK1 and ERK2 cascade*”, “*positive regulation of pathway-restricted SMAD protein phosphorylation*”, “*cellular response to gonadotropin stimulus*”, “*cellular response to hormone stimulus*”, “*response to steroid hormone*”, “*ovulation cycle*”, “*response to estradiol*”, “*MAPK cascade*” and “*positive regulation of MAPK cascade*” (Table 2). The EF ranged between 2.0 for “*cellular response to hormone stimulus*” to8.3 for “*cellular response to FSH stimulus*” (Table 2).

Other biological processes potentially related to HPGA, shown in Table 2, were also enriched with *p* values < 0.05 but had FDR values higher than 0.05. These biological processes were “*prostate gland stromal morphogenesis*” (GO:0060741) (EF = 14, *p*-value = 1.9 × 10^−2^ and FDR = 2.5 × 10^−1^); “*cellular response to luteinizing hormone stimulus*” (GO:0071373) (EF = 6.9, *p*-value = 4.9 × 10^−2^ and FDR = 4.5 × 10^−1^) and “*positive regulation of ovulation*” (GO:0060279) (EF = 14, *p*-value = 1.9 × 10^−2^ and FDR = 2.5 × 10^−1^).

### 2.4. Effect of Cd/Se-QD-NPs on Hypothalamic-Pituitary-Gonadal Axis in T98G Human Glioblastoma Cells

Fuster and co-workers [19] exposed T98G human glioblastoma cells for 48 h to 40 µg/mL of Cd/Se-QDs-NPs. This non-cytotoxic treatment dysregulated the expression of 544 genes, 31 of them had a FC lower than 0.5, 31 genes had a FC higher than 2 and 482 genes had an intermediate FC between 0.5 and 2. We uploaded the complete list of the DEGs (taken from [19]) to the GO resource, and a performed statistical overrepresentation test using PANTHER pathways and GO biological processes as annotation datasets. The results are summarized in Table 1 and Table 3.

The GnRH receptor pathway was overrepresented among the DEGs after the Cd/Se-QDs-NPs exposure because 15 of the 237 genes were altered, while only six of those were expected to be altered. The EF was 2.5 (Table 1).

Five different biological processes, potentially involved in HPGA regulation, were overrepresented among the set of DEGs after exposure to Cd/Se-QDs-NPs. These biological processes were “*regulation of ERK1 and ERK2 cascade*”, “*response to estradiol*”, “*response to hormone*”, “*regulation of MAPK cascade*” and “*regulation of hormone secretion*” (Table 3). The EF ranged between 2.0 for “*response to hormone*” and 3.6 for “*response to estradiol*” (Table 3).

In addition to the biological processes shown in Table 3, others also potentially related to the HPGA were enriched with *p* values < 0.05, although these cases did not exceed the threshold for false-positives and had an FDR > 0.05. These biological processes were: “*positive regulation of progesterone biosynthetic process*” (GO:2000184) (EF = 40, *p*-value = 4.9 × 10^−2^ and FDR = 5.5 × 10^−1^); “*regulation of gonadotropin secretion*” (GO:0032276) (EF = 7; *p*-value = 3.9 × 10^−2^ and FDR = 5.1 × 10^−1^); “*prostate gland stromal morphogenesis*” (GO:0060741) (EF = 27, *p*-value = 5.7 × 10^−3^ and FDR = 1.7 × 10^−1^); “*positive regulation of FSH secretion*” (GO:0046881) (EF = 13, *p*-value = 1.5 × 10^−2^ and FDR = 3.0 × 10^−1^); “*regulation of gonadotropin secretion*” (GO:0032276) (EF = 7, *p*-value = 3.9 × 10^−2^, FDR = 5.3 × 10^−1^); “*positive regulation of gonadotropin secretion*” (GO:0032278) (EF = 11, *p*-value = 1.9 × 10^−2^ and FDR = 3.4 × 10^−1^) and “*negative regulation of hormone secretion*” (GO:0046888) (EF = 4.2, *p*-value = 2.2 × 10^−3^ and FDR = 8.3 × 10^−2^).

### 2.5. Effect of Ag-NPs on Hypothalamic-Pituitary-Gonadal Axis in T98G Human Glioblastoma Cells

Fuster and co-workers [18] exposed T98G human glioblastoma cells for 48 h to 40 µg/mL of Ag-NPs. This non-cytotoxic treatment dysregulated the expression of 43 genes, 13 of which were up-regulated and the remaining 30 were down-regulated. We uploaded this list of DEGs (taken from [18]) to the GO resource and analyzed the alterations in the pathways. The results are summarized in Table 1 and Table 4.

The oxidative stress response was the only altered PANTHER pathway among the 43 genes dysregulated in the T98G cells after exposure to Ag-NPs. An overrepresentation of the GnRH hormone receptor pathway occurred among the 43 DEGs, with three genes belonging to this pathway out of a total of 237 genes. The EF in this case was 6.4 and the *p* value was 1.2 × 10^−2^, although the threshold for false positives was exceeded because the FDR was 4.8 × 10^−1^.

Six different biological processes, potentially involved in HPGA regulation, were enriched among the set of DEGs after exposure to Ag-NPs. These biological processes were “*negative regulation of ERK1 and ERK2 cascade*”, “*positive regulation of ERK1 and ERK2 cascade*”, “*negative regulation of MAPK cascade*”, “*negative regulation of MAP kinase activity*”, “*MAPK cascade*” and “*response to hormone*” (Table 4). The EF ranged between 5.1 for “*response to hormone”* and 26 for “*Negative regulation of MAP kinase activity*” (Table 4). In addition to these six biological processes, “*response to estradiol*” (GO:0032355) was also reported with a *p* value < 0.05 (2.0 × 10^−3^), and with an FDR of 1.3 × 10^−1^. In this case, the EF was 12.

### 2.6. Genes of the GnRH Hormone Receptor Pathway Altered after Exposure to Several NPs

As previously mentioned, the GnRH receptor pathway (Figure 2, Appendix A) is operated by 237 genes, which are listed in Appendix A. The number of DEGs in the T98G cells after exposure to TiO_2_-, Cd/Se-QDs- and Ag-NPs was 28, 15 and 3, respectively (Table 1). These genes, together with their respective FC, are listed in Table 5. Figure 3 shows a Venn diagram with the proportions of overlapped DEGs among the three NPs.

Two of the three dysregulated genes after Ag-NPs exposure were also dysregulated after exposure to TiO_2_-NPs and Cd/Se-QDs-NPs. These genes were Jun B proto-oncogene, which was down-regulated in all three cases by around 58–52%, and early growth response 1, which was down-regulated by around 80% for TiO_2_- and Cd/Se-QDs-NPs and even more (by around 93%) in the case of silver) (Table 5). The third dysregulated gene (up-regulated by around 30%) after exposure to Ag-NPs (EGF receptor) was also up-regulated by around 60% after exposure to TiO_2_-NPs (Table 5). Nine of the 28 genes dysregulated by exposure to TiO_2_-NPs were also dysregulated after exposure to Cd/Se-QDs-NPs (Table 5, Figure 3). Six of the 15 genes altered after exposure to the Cd/Se-QDs-NPs, but were not altered after exposure to the other two NPs and 18 out of the 28 DEGs after exposure to TiO_2_-NPs, were not noted differentially expressed after exposure to Ag- and Cd/Se-QDs-NPs (Figure 3). The gene with the highest up-regulation level (2.52-fold) was heat shock protein family A (Hsp70) member 1A, which was reported after exposure to TiO_2_-NPs exposure. The gene with the highest down-regulation level (0.074-fold) was early growth response 1, reported after exposure to Ag-NPs.

## 3. Discussion

We reassessed our previous data by studying the effects on the transcriptome of the T98G human glioblastoma cells after exposure to metallic NPs. We previously found that the exposure of T98G cells to Ag-, TiO_2_- and Cd/Se-QDs-NPs, but not to ZnO-NPs, was able to alter a number of biological pathways and molecular processes related mainly to neuroinflammation and blood-brain barrier integrity [17,18,19]. In this previously published analysis, we noted potential alterations to the GnRH receptor pathway, but did not realize the potential importance of this finding. Here we reassessed the information and noted that these alterations in the GnRH receptor pathway, and other biological processes related to the HPGA, is a repetitive observation in several NPs, and therefore, it could indeed be relevant for the hazard identification for fertility of NPs. Our first analysis noted that of the 1025 DEGs after exposure to TiO_2_-NPs, 817 were for alterations of either an FC higher than 0.5 or an up-regulations lower than 2 and these genes with moderate dysregulation caused background noise. So, we decided to not take them into account [17]. Likewise, and for similar reasons, 482 of a total of 544 genes, altered after exposure to Cd/Se-QDs-NPs, were ruled out of the analysis [19]. In this paper, we used the whole DEGs data set, which allowed for the identification of a number of biological processes related to alterations in the HPGA that were skipped in our previous analysis and demonstrated the relevance of the evidence that NPs could act in an endocrine disruption-based mode of action.

### 3.1. Alterations to the GnRH Pathway

The GnRH receptor transmits signals via several key components as MAPK-cascades [22,23,24] (Figure 2). Table 2, Table 3 and Table 4 show for the three studied NPs counts of several MAPK cascades with a number of overrepresented genes among those DEGs after treatment with TiO_2_-NPs (Table 2) (“*MAPK cascade*” and “*positive regulation of MAPK cascade*”); Cd/Se-QDs- NPs (Table 3) (“*regulation of MAPK cascade*”) and Ag-NPs (Table 4) (“*negative regulation of MAPK cascade*”, “*MAPK cascade*” and “*negative regulation of MAP kinase activity*”).

In addition to the MAPK cascades, other kinases like ERKs also play a critical role downstream of the binding of GnRH to its receptor [22,23,24] (Figure 2). These ERKs were also overrepresented on the list of DEGs after exposure to NPs. Specifically, “*regulation of ERK1 and ERK2 cascade*” after exposure to Cd/Se-QDs-NPs (Table 3), “*negative regulation of ERK1 and ERK2 cascade*” after exposure to TiO_2_-NPs (Table 2), “*negative regulation of ERK1 and ERK2 cascade*” and “*positive regulation of ERK1 and ERK2 cascade*” after exposure to Ag-NPs (Table 4).

All this information generally suggests that the exposure to these metallic NPs causes alterations to the GnRH pathway. The binding of GnRH to its receptor triggers HPGA with the further release of LH and FSH, which act on gonads by causing the release of male and female sex hormones (Figure 1). Hence the GnRH receptor pathway alteration should impact the HPGA.

### 3.2. Alterations to the HPGA

Several biological processes that address the HPGA were overrepresented from among the list of DEGs after exposure to NPs. This is the case for the overrepresentations of biological processes “*cellular response to FSH stimulus*”, “*cellular response to gonadotropin stimulus*”, “*cellular response to hormone stimulus*”, “*response to steroid hormone*”, “*ovulation cycle*” and “*response to estradiol*” noted after exposure to TiO_2_-NPs (Table 2); “*response to estradiol*”, “*response to hormone*” and “*regulation of hormone secretion*” noted after exposure to Cd/Se-QDs-NPs (Table 3) and “*response to hormone*” noted after exposure to Ag-NPs (Table 4).

Other biological processes related to the HPGA were also overrepresented, such as “*prostate gland stromal morphogenesis*”, “*cellular response to luteinizing hormone stimulus*”, “*positive regulation of ovulation*” after exposure to TiO_2_-NPs; “*positive regulation of progesterone biosynthetic process*”, “*regulation of gonadotropin secretion*”, “*prostate gland stromal morphogenesis*”, “*positive regulation of FSH secretion*”, “*regulation of gonadotropin secretion*”, “*positive regulation of gonadotropin secretion*” and “*negative regulation of hormone secretion*” after exposure to Cd/Se-QDs-NPs and “*response to estradiol*” after exposure to Ag-NPs. In all the above cases, overrepresentation was statistically significant (*p* value lower than 0.05), although the 5% threshold for false positives was exceeded. However, it was noted that, when considering the information in Table 2, Table 3 and Table 4 (all data was supported with statistical significance), it would be unlikely that these cases were false positives generated by chance and indeed can be considered as supporting information of the data shown in Table 2, Table 3 and Table 4.

### 3.3. Regulation of the Hypothalamic-Pituitary-Gonadal Axis by Glial Cells

The results presented herein were obtained using T98G cells, a multiformal human glioblastoma and not in neurons that could secrete GnRH. It should be noted that GnRH neurons do not express the male and female steroid hormone receptors needed for the feedback regulation of the HPGA (Figure 2) [25]. Thus, the action of other cells in HPGA regulation is needed. These other cells are glial cells, as we used in our previous experiments [17,18,19]. Indeed, these cells have been suspected of causing infertility disorders, such as polycystic ovary syndrome [25]. It is suggested that glia is able to release soluble factors as transforming factors and neurorregulins, which may affect neurons and other glial cells for regulating the GnRH pathway [26].

The literature proposes that the glial cells of median eminences regulate GnRH secretion via mechanisms that are based on the release of growth factors, which act on receptors that activate tyrosine kinase activity downstream and other mechanisms that involve plastic rearrangements of adhesion between neurons and glia [27,28]. At this point, it is necessary to point out that ERK and MAPK, which are also tyrosine-kinases, were severely affected (Table 2, Table 3 and Table 4) by exposure to the metallic NPs considered in this study. This supports the hypothesis that these alterations to MAPK and ERK could indeed induce alterations in glia’s role of regulating GnRH neurons and, consequently, impair the HPGA performance.

It is believed that some transcription growth factors of the β family (TGF-β) are released from astrocytes to modulate the release of GnRH and, consequently, steroids according to the scheme shown in Figure 1 for the working of the HPGA [28]. These TGF-β operate via receptors that use SMAD proteins as intracellular transducers [28]. It is noted that the T98G cells are a multiformal glioblastoma that contain astrocytes, and therefore, also the genes for the biological process of “*positive regulation of pathway-restricted SMAD protein phosphorylation*”, which were overrepresented among those DEGs after exposure to TiO_2_-NPs, as well as the genes belonging to the TGF-β signaling pathway. These facts support the hypothesis that NPs disturb the release of GnRH and, consequently, all the downstream events of the HPGA.

The GnRH receptor is coupled to G-proteins, which exert a downstream effect on adenylate cyclase (Figure 2) [23]. Table 5 shows how adenylate cyclase was dysregulated 1.9-fold after exposure to both TiO_2_-NPs and Cd/Se-QD-NPs. Once again, this suggests that the HPGA could be altered, which could be related to the mechanism of potential fertility impairments. Another alteration in the expression of genes that suggested this same hypothesis was the EGF receptor (Figure 2), which was up-regulated after exposure to Ag- and TiO_2_-NPs (Table 5).

## 4. Materials and Methods

We reevaluated our previously generated RNAseq data. The T98G human glioblastoma cells were exposed to 40 µg/mL of Ag-NPs, 20 µg/mL of TiO_2_-NPs, 5 µg/mL of ZnO-NPs and 40 µg/mL of Cd/Se-QDs-NPs for 72 h. The RNA was further isolated using the Trizol reagent according to the manufacturer’s instructions and all samples with an RNA integrity number greater than 7 were sent to an external provider (MACROGEN Inc., Seoul, Public of Korean (https://dna.macrogen.com/ (accessed on 10 July 2023))) for subsequent RNAseq experiments on the Illumina platform using paired-end 101-bp reads [17,18,19]. The raw data from these RNAseq experiments are available in the Sequence Read Archive database (https://www.ncbi.nlm.nih.gov/sra/ (accessed on 10 July 2023)) under accession number SAMN13151876. The bioinformatic analysis was performed as reported by Fuster and coworkers [17,18,19]. The following software was used for this analysis: (i) Trimmomatic v. 0.36 to trim low quality bases and all the remaining bases and any remaining adapter sequences present in the reads, (ii) Hisat2 v. 2.1.0 to map the read pairs to the GRCh38 version of the human reference genome, (iii) Samtools v. 0.1.19 to compress the resulting SAM files into BAM files and (iv) Cufflinks v. 2.2.1 to quantify the gene expression levels and perform the statistical comparisons.

The physical-chemical properties of these NPs are provided in Table 6.

The T98G human glioblastoma cells were obtained from the European Collection of Authenticated Cell Cultures (UK) (catalogue no. 92090213). The T98G cells were cultured following the manufacturer’s instructions. The cell culture medium was Dulbecco’s modified Eagle Medium-GlutaMAX™-I with glucose and sodium pyruvate supplemented with fetal bovine serum and non-essential amino acids, penicillin and streptomycin. The environmental conditions of the cell culture were 5% CO_2_ and 37 °C.

In this study, we uploaded the DEGs to the GO Resource (released on 10 May 2023) (http://geneontology.org/) (accessed on 10 July 2023) [29,30] and we analyzed the data for the PANTHER pathway annotation dataset and for the biological process annotation dataset. In line with the main scope of this manuscript, we centered our assessment on the pathways related to the HPGA (Figure 1). The statistical significance of the results was determined by Fisher’s Exact test and the FDR was used to control excess false positives.

The Venn diagrams were created with Venny 2.1 (available at https://bioinfogp.cnb.csic.es/tools/venny/ (accessed on 10 July 2023) [31].

## 5. Conclusions

The data provided in this manuscript assess the possibility that NPs could cause alterations that could be related with fertility impairments through a mode of action based on endocrine disruption, specifically by disrupting HPGA at the GnRH receptor level. This opens up a new field to study to investigate the possible deleterious effects of NPs on fertility via an endocrine disruption mechanism that, to the best of our knowledge, has not yet been considered.

The RNAseq experiments that generated the data assessed in this paper were conducted under conditions of low cytotoxicity exposure (the viability of T98G cells after exposure was higher than 90% in all cases). Perhaps these experimental conditions were not entirely appropriate to identify the hazard being addressed in this study. Therefore, further specific and targeted research is needed to verify this hypothesis and examine the real risk that this endocrine disruption mode of action may pose. Nevertheless, the existence of this endocrine disruption-based mode of action does not exclude the possibility that NPs could exert fertility impairments through other modes of action, such as oxidative stress, which has already been reported by several authors.

## Figures and Tables

**Figure 1 ijms-24-11687-f001:**
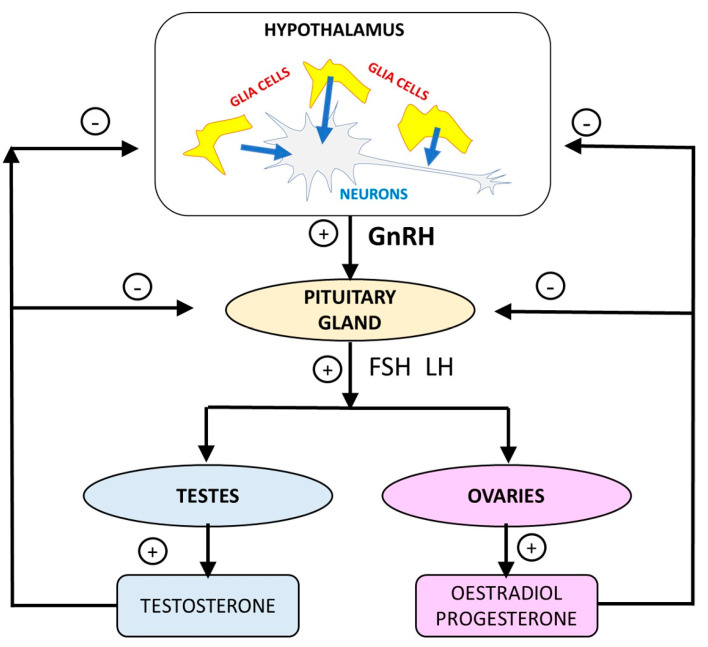
Hypothalamic-pituitary-gonadal axis. Glial cells regulate gonadotropin releasing hormone neurons. GnRH = Gonadotropin-releasing hormone. LH = Luteinizing hormone. FSH = Follicle-stimulating hormone. + = Positive feedback (hormone release). − = Negative feedback (inhibition of hormone release).

**Figure 2 ijms-24-11687-f002:**
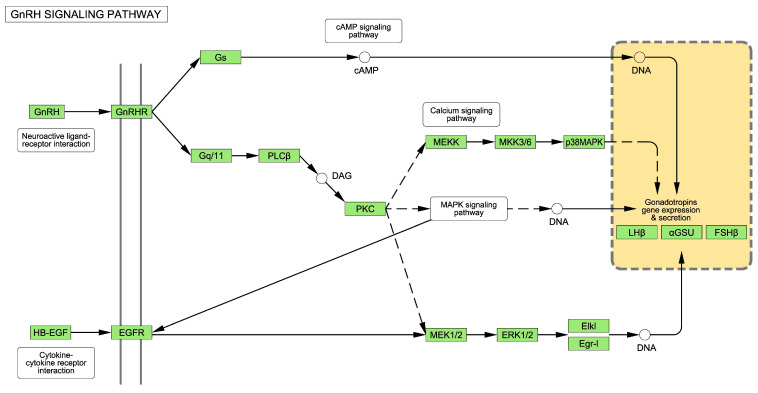
Simplified GnRH signaling pathway for *Homo sapiens*. The Figure was modified starting from the whole pathway provided by KEGG (hsa04912). The whole pathway is provided in Supplementary Material. GnRH = Gonadotropin releasing hormone. GnRHR = Gonadotropin releasing hormone receptor. Gs and Gq/11 = G-proteins. PLC = Phospholipase C. DAG = Diacyl-glycerol. PKC = Protein Kinase C. MEKK, MKK3/6, p38MAPK, MAPK = different mitogen activates protein kinases. HB-EGF = Heparin binding epidermal growth factor. EGFR = Epidermal growth factor receptor. MEK. Mitogen-activated ERK kinase. ERK1/2 = extracellular signal-regulated kinases 1 and 2. Elk1 = Transcription factor ELK. Egr-1 = Early growth response protein 1. LH = Luteinizing hormone. FSH = Follicle-stimulating hormone. GSU = Glycoprotein hormone alpha-subunit.

**Figure 3 ijms-24-11687-f003:**
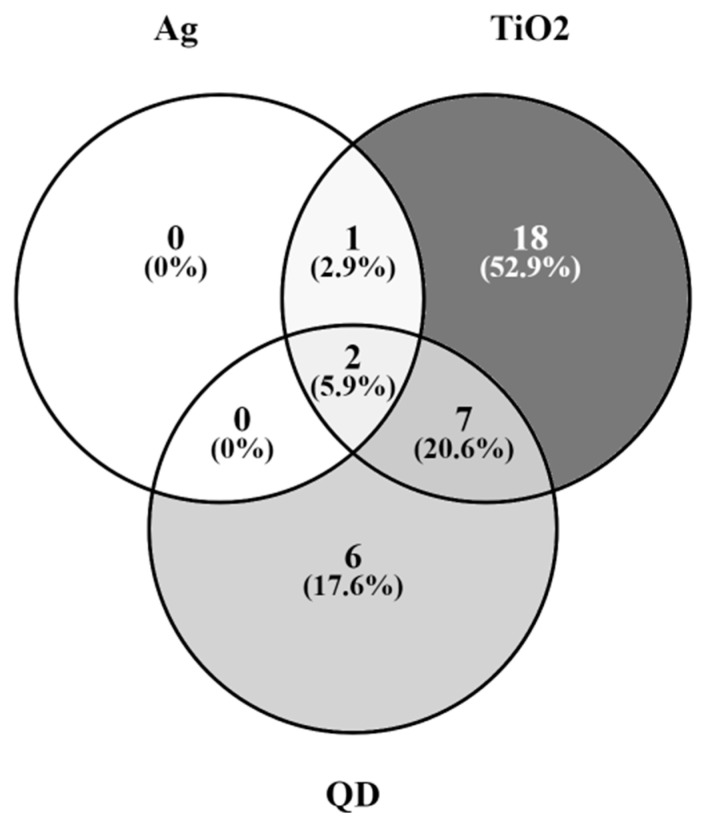
Venn diagram of DEGs in the T98G cells exposed to TiO_2_-, Cd/Se-QDs- and Ag-NPs. Venn diagram built from the data shown in Table 5 and based on the data provided by [17,18,19].

**Table 1 ijms-24-11687-t001:** Overrepresentation of the GnRH pathway in the differentially expressed genes in the T98G human glioblastoma cells exposed to different NPs. Lists of DEGs were taken from [17,18,19] and reassessed using the PANTHER annotation system. DEGs = differentially expressed genes. EF = Enrichment factor. Only the pathways with both a *p* value and an FDR < 0.05 are shown.

PANTHER Pathway	Total Genes in the Pathway	DEGs after NP Exposure	Expected among DEGs	EF	*p* Value	FDR
TiO_2_-NPs	237	28	11	2.5	4.25 × 10^−5^	2.27 × 10^−3^
Cd/Se-QDs-NPs	237	15	6	2.5	1.5 × 10^−3^	4.7 × 10^−2^
Ag-NPs	237	3	0.5	6	1.2 × 10^−2^	4.8 × 10^−1^

**Table 2 ijms-24-11687-t002:** Overrepresentation of the biological process annotations in the differentially expressed genes in the T98G human glioblastoma cells exposed to TiO_2_-NPs. The complete set of DEGs from [17] was reassessed in the GO biological process annotation. DEGs = differentially expressed genes. EF = Enrichment factor. Only the biological processes with both a *p* value and an FDR < 0.05 are shown.

Biological Process	Total Genes Biological Process	DEGs after TiO_2_-NP Exposure	Expected among DEGs	EF	*p* Value	FDR
Cellular response to FSH stimulus (GO:0071372)	12	5	0.6	8.3	8.0 × 10^−4^	2.3 × 10^−2^
Negative regulation of ERK1 and ERK2 cascade (GO:0070373)	71	12	3.4	3.5	4.0 × 10^−4^	1.4 × 10^−2^
Positive regulation of pathway-restricted SMAD protein phosphorylation (GO:0060394)	50	9	2.4	3.7	1.4 × 10^−3^	3.6 × 10^−2^
Cellular response to gonadotropin stimulus (GO:0071371)	20	6	1	6.0	9.7 × 10^−4^	2.7× 10^−2^
Cellular response to hormone stimulus (GO:0032870)	509	50	25	2.0	7.9 × 10^−6^	5.0 × 10^−4^
Response to steroid hormone (GO:0048545)	289	34	14	2.4	1.0 × 10^−5^	6.3 × 10^−4^
Ovulation cycle (GO:0042698)	75	12	3.6	3.3	6.2 × 10^−4^	1.9 × 10^−2^
Response to estradiol (GO:0032355)	121	15	5.8	2.6	1.5 × 10^−3^	3.8 × 10^−2^
MAPK cascade (GO:0000165)	216	24	10	2.4	4.6 × 10^−4^	1.5 × 10^−2^
Positive regulation of MAPK cascade (GO:0043410)	483	48	23	2.1	8.1 × 10^−6^	5.1 × 10^−4^

**Table 3 ijms-24-11687-t003:** Overrepresentation of the biological process annotations in the differentially expressed genes in the T98G human glioblastoma cells exposed to Cd/Se-QDs-NPs. The complete set of DEGs [19] was reassessed in the GO biological process annotation. DEGs = differentially expressed genes. EF = Enrichment factor. Only the biological processes with both a *p* value and an FDR < 0.05 are shown.

Biological Process	Total Genes Biological Process	DEGs after Cd/Se-QDs Exposure	Expected among DEGs	EF	*p* Value	FDR
Regulation of ERK1 and ERK2 cascade (GO:0070372)	301	21	7.6	2.8	5.2 × 10^−5^	4.4 × 10^−3^
Response to estradiol (GO:0032355)	121	11	3.0	3.6	4.1 × 10^−4^	2.3 × 10^−2^
Response to hormone (GO:0009725)	789	39	20	2.0	1.2 × 10^−4^	8.8 × 10^−3^
Regulation of MAPK cascade (GO:0043408)	672	35	17	2.1	9.5 × 10^−5^	8.0 × 10^−3^
Regulation of hormone secretion (GO:0046883)	253	16	6.4	2.5	1.1 × 10^−3^	4.7 × 10^−2^

**Table 4 ijms-24-11687-t004:** Overrepresentation of the biological process annotation in the differentially expressed genes in the T98G human glioblastoma cells exposed to Ag-NPs. The complete set of DEGs from [18] was reassessed in the GO biological process annotation. DEGs = differentially expressed genes. EF = Enrichment factor. Only the biological processes with both a *p* value and an FDR < 0.05 are shown.

Biological Process	Total Genes Biological Process	DEGs after Cd/Se-QDs Exposure	Expected among DEGs	EF	*p* Value	FDR
Negative regulation of ERK1 and ERK2 cascade (GO:0070373)	71	3	0.14	21	4.3 × 10^−4^	4.3 × 10^−2^
Positive regulation of ERK1 and ERK2 cascade (GO:0070374)	215	5	0.43	12	7.2 × 10^−5^	1.2 × 10^−2^
Negative regulation of MAPK cascade (GO:0043409)	171	5	0.34	15	2.5 × 10^−5^	5.1 × 10^−3^
Negative regulation of MAP kinase activity (GO:0043407)	58	3	0.12	26	2.4 × 10^−4^	3.0 × 10^−2^
MAPK cascade (GO:0000165)	216	6	0.43	14	4.8 × 10^−6^	1.9 × 10^−3^
Response to hormone (GO:0009725)	789	8	1.6	5.1	1.5 × 10^−4^	2.0 × 10^−2^

**Table 5 ijms-24-11687-t005:** Genes of the gonadotropin-releasing hormone receptor pathway with a statistically altered expression after exposure to different NPs. The table was built by transforming the raw data provided by references [15,16,17]. FC = fold change.

		FC		
Gene	Name	TiO_2_	Cd/Se-QDs	Ag
ENSG00000131791	Protein kinase AMP-activated non-catalytic subunit beta 2 (PRKAB2)	1.35	-	-
ENSG00000185950	Insulin receptor substrate 2 (IRS2)	1.45	0.77	-
ENSG00000164111	Annexin A5 (ANXA5)	1.30	-	-
ENSG00000204388	Heat shock protein family A (Hsp70) member 1B (HSPA1B)	2.44	-	-
ENSG00000107968	Mitogen-activated protein kinase kinase kinase 8 (MAP3K8)	0.56	0.59	-
ENSG00000104899	Anti-Mullerian hormone (AMH)	0.50	-	-
ENSG00000164742	Adenylate cyclase 1 (ADCY1)	1.88	1.61	-
ENSG00000095015	Mitogen-activated protein kinase kinase kinase 1 (MAP3K1)	0.73	-	-
ENSG00000204389	Heat shock protein family A (Hsp70) member 1A (HSPA1A)	2.52	-	-
ENSG00000096433	Inositol 1,4,5-trisphosphate receptor type 3 (ITPR3)	1.43	-	-
ENSG00000162734	Proliferation and apoptosis adaptor protein 15 (PEA15)	1.62	-	-
ENSG00000106617	Protein kinase AMP-activated non-catalytic subunit gamma 2 (PRKAG2)	1.57	-	-
ENSG00000168610	Signal transducer and activator of transcription 3 (STAT3)	0.73	-	-
ENSG00000177606	Jun proto-oncogene, AP-1 transcription factor subunit (JUN)	1.34	-	-
ENSG00000171223	Jun B proto-oncogene	0.42	0.45	0.48
ENSG00000117394	Solute carrier family 2 member 1 (SLC2A1)	1.31	-	-
ENSG00000167552	Tubulin alpha 1a (TUBA1A)	1.30	1.25	-
ENSG00000117318	Inhibitor of DNA binding 3 (ID3)	0.67	0.48	-
ENSG00000069011	Paired like homeodomain 1 (PITX1)	0.52	-	-
ENSG00000122641	Inhibin subunit beta A (INHBA)	2.20	-	-
ENSG00000162772	Activating transcription factor 3 (ATF3)	2.16	0.73	-
ENSG00000169083	Androgen receptor (AR)	0.77	-	-
ENSG00000171951	Secretogranin II (SCG2)	0.73	-	-
ENSG00000146648	Epidermal growth factor receptor (EGFR)	1.61	-	1.26
ENSG00000170345	Fos proto-oncogene	0.31	0.22	-
ENSG00000100968	Nuclear factor of activated T cells 4 (NFATC4)	0.48	-	-
ENSG00000122420	Prostaglandin F receptor (PTGFR)	0.77	-	-
ENSG00000120738	Early growth response 1 (EGR1)	0.18	0.19	0.074
ENSG00000123416	Tubulin alpha 1b (TUBA1B)	-	1.42	-
ENSG00000134909	Rho GTPase activating protein 32 (ARHGAP32)	-	1.52	-
ENSG00000149295	Dopamine receptor D2 (DRD2)	-	2.10	-
ENSG00000166949	SMAD family member 3 (SMAD3)	-	1.39	-
ENSG00000163083	Inhibin subunit beta B (INHBB)	-	1.82	-
ENSG00000130522	Jun D proto-oncogene	-	1.64	-

**Table 6 ijms-24-11687-t006:** Main physical-chemical properties of the NPs analyzed in this manuscript. Size was determined by transmission electronic microscopy. Data taken from [15,16,17].

NP	SIZE (nm)	Z-Potential (mV)
Ag	17 ± 9	−35 ± 1.5
TiO_2_	18 ± 5	23 ± 0.8
ZnO	23 ± 9	17 ± 0.6
Cd/Se-QDs	4 ± 0.6	−32 ± 1

## Data Availability

The raw data of these RNAseq experiments are available in the Sequence Read Archive database (https://www.ncbi.nlm.nih.gov/sra/ (accessed on 10 July 2023) with accession number SAMN13151876.

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
