# Peer review of "Investigation of the Effects of Metallic Nanoparticles on Fertility Outcomes and Endocrine Modification of the Hypothalamic-Pituitary-Gonadal Axis"

_ijms, 2023, doi:10.3390/ijms241411687_

Round 1

Reviewer 1 Report

The authors are concerned about nanotechnology and human exposure to nanoparticles, and in particular the influence on fertility.  They reviewed published RNAseq studies by looking at effects of some nanoparticles on the transcriptome of T98G human glioblastoma cells.  This is a cell line derived from a human male.  The authors “reassessed” their previously derived data (Lines 231-233), thus this is not an experimental study.

 The utility of the cell model needs explanation and rationale in the Introduction.

 The authors provide no clear hypothesis.  See lines 93-99.  This paper might be better conceived as a “perspective piece” or some other retrospective.

 The authors cited their work five times.

 The choice of cell type is not well described to their proposed endpoint of fertility.  If the endpoint was HPGA axis disruption, that might be more relevant to the review, as there are other mechanisms beyond that biological hierarchy that results in fertility or not. 

 Fig. 1 is very nice.

Fig. 2 legend needs to cite the source of where from which it was reproduced (15).  The figure is not useful because it does not include names for all acronyms.  If they were in the legend, it is still too busy to be easy for the reader.

Moderate English language adjustments are encouraged.

 Line 118.  Please clarify if cells were reduced to 10% viability or by 10% and cite the paper (11)?

 Overall, the paper has good intent, but is a bit presumptuous and could be made more objective.  Another format for the content and some condensation of the content would be useful.  There is no direct line from their cell line choice to gene expressions to fertility.

Needs help.

Author Response

ANSWERS FOR REVIEWER 1

General comment from authors:

Thank you very much to the reviewer by his/her comments because after implementation of needed changes the manuscript will be notably improved.

We would like first to state a clarification about the origin and conception of this manuscript. This is indeed a retrospective manuscript as the reviewer commented in the sense that we are considering previously published data. We performed a RNAseq experiments with a cell line of human glioblastoma with the aim of studying molecular mechanism of neurotoxicity induced by nanomaterials. This is the reason because these results are shown in this cell model. Afterwards, we noted in our data certain repetitive results about the alterations in transcriptome of gonadotropin releasing hormone receptor pathway. This is when we decided to take a more in deep insight on this data modifying certain parameters in our previous bioinformatic analysis to focus and address this issue. The validity of the model was given by the fat that it is described that glia plays a pivotal role in the regulation of the gonadotropin releasing neurons in the hypothalamus. Thus, whether transcriptome of glia is altered by exposure to nanomaterials the regulation of gonadotropin releasing neurons could be altered too. In order to present in the manuscript a more clear and consistent clarification about the validity of the cell model we have added in the introduction (see lines 105-120 in the current manuscript) this clarification and modified Figure 1 accordingly.

We are aware that our hypothesis that nanoparticles can alter hypothalamic-pituitary-gonadal axis should be further verified in other experiments and with other cellular models. However, to our knowledge this is the first report alerting that nanoparticle can impair fertility via an endocrine disruption mode of action; which could also contribute to clarify the controversial results about whether nanoparticles are able to cause alterations in fertility or not. 

Reviewer comment: The authors are concerned about nanotechnology and human exposure to nanoparticles, and in particular the influence on fertility.  They reviewed published RNAseq studies by looking at effects of some nanoparticles on the transcriptome of T98G human glioblastoma cells.  This is a cell line derived from a human male.  The authors “reassessed” their previously derived data (Lines 231-233), thus this is not an experimental study.

 The utility of the cell model needs explanation and rationale in the Introduction.

Authors’ answer: Please, see above our general comment.

Reviewer comment: The authors provide no clear hypothesis.  See lines 93-99.  This paper might be better conceived as a “perspective piece” or some other retrospective.

Authors’ answer: Please, see above our general comment.

Reviewer comment: The authors cited their work five times.

Authors’ answer: We have used four cites of our own authorship. Three of them are the cites in which the original data is shown and we consider absolutely necessary to describe the original data that are being analyzed. It is true that these three papers are cited several times along the manuscript in the different subsections of the work. The fourth cite is a supporting reference and is mentioned only once.

Reviewer comment: The choice of cell type is not well described to their proposed endpoint of fertility.  If the endpoint was HPGA axis disruption, that might be more relevant to the review, as there are other mechanisms beyond that biological hierarchy that results in fertility or not. 

Authors’ answer: Please, see our comments above as regard the choice of the cell model. As we commented above, we are aware that other experiments with other cell lines will be needed in which other hierarchies would be explored. However, we consider in this case that we cannot go beyond what our data shown we prefer to keep the manuscript circumscribed to the hypothalamic-pituitary-gonadal axis.

Reviewer comment: Fig. 1 is very nice.

Authors’ answer: Thank you very much. As commented above we have slightly modified the figure to introduce the role of glia cells in the hypothalamic-pituitary-gonadal axis regulation.

Reviewer comment: Fig. 2 legend needs to cite the source of where from which it was reproduced (15).  The figure is not useful because it does not include names for all acronyms.  If they were in the legend, it is still too busy to be easy for the reader.

Authors’ answer: In the previous version, the Figure 2 was the original full pathway published by Kyoto Encyclopedia of Genes and Genomes database and we are reproducing this Figure with permission. We note after the reviewer’s comment that could be indeed too complex. We have simplified and modified the Figure showing exclusively all pathways for which we find alterations in our data. The original full Figure has been moved to Supplementary Material for consultation of readers if needed.

Reviewer comment: Moderate English language adjustments are encouraged.

Authors’ answer: The manuscript has been reviewed by a professional scientific English translator that has introduced several gramma corrections.

Reviewer comment: Line 118.  Please clarify if cells were reduced to 10% viability or by 10% and cite the paper (11)?

Authors’ answer: We have clarified the text. The experiments were performed after exposures reducing cell viability by 10% (90% of cell viability remain after three days of exposure).

Reviewer comment: Overall, the paper has good intent, but is a bit presumptuous and could be made more objective.  Another format for the content and some condensation of the content would be useful.  There is no direct line from their cell line choice to gene expressions to fertility.

Authors’ answer: Please, see our comments at the beginning about how this manuscript was conceived and the reason of the cell line. In order to avoid the possibility to go beyond what our showed data display we have modified the title by the new one of: “Endocrine disruption. A possible mode of action to explore in the future for elucidating whether metallic nanoparticles can cause fertility alterations”.

We have also modified the conclusions accordingly.

Reviewer 2 Report

Review of the manuscript entitled: Can Some Metallic Nanomaterials Cause Fertility Impairments Via an Endocrine Disruption-Based Mode of Action? The manuscript is interesting. Moreover, in my opinion the manuscript is prepared very well but some corrections should be made.

1.      In abstract and introduction clear aim of the manuscript should be added e.g. "The aim of the present study was to ...". In “abstract”, aim should be at the beginning or after a short introduction and in “introduction”, aim should be at the end.

2.      Maybe interesting for readers will be some information regarding the removal of NPs from the body. Is there such information in the literature?

3.      Please improve the quality of Fig. 2 similar Fig. 3.

4.      In my opinion the subsections in the discussion should be removed. The discussion should be a single text.

5.      Material and methods must include the exact cell culture conditions, reagents used and their catalog numbers. The entire material and methods section should be revised. The lack of this data is disrespectful to reviewers and readers.

Author Response

(The authors gave the same response as above.)

Round 2

Reviewer 1 Report

See the attachment.

Some editing is still needed.

Author Response

We would like to thank the reviewer for his efforts in this new round of additional comments, which we have tried to implement in this new submission.

Reviewer’s comment: The manuscript is vastly improved. Please find some suggestions. The manuscript wording could use some more tightening up.

Authors’ response: Thank you to the reviewer for recognizing our efforts to improve the manuscript.

Reviewer’s comment: Title. Too long. How about being more specific? E.g., “Investigation into endocrine modification to the HPG axis by metallic nanoparticles and fertility outcomes”

Authors’ response: To avoid non-standard abbreviations in the title, but following the suggestion, we have adopted the following title: “Investigating endocrine modification of the hypothalamic-pituitary-gonadal axis by metallic nanoparticles and fertility outcomes

Reviewer’s comment: Many people define fertility in different ways. Are you meaning by offspring produced? By conception? By sperm quality? Please define it up front.

Authors’ response: We use fertility in this paper in the same sense as OECD TG 421 (Reproduction/Developmental Toxicity Screening Test). We have now introduced this consideration in lines 141-143 where there is a sentence that reads: “Overall, the aim of the present study is to provide knowledge to elucidate whether metallic NPs can induce fertility disorders altering male or female reproductive functions in aspects such as conception, sperm quality, gonadal impairments, alterations in female reproductive cycle, and other usually evaluated, for example, in OECD TG 421.”  

Reviewer’s comment: Glia should be glial when used as an adjective.

Authors’ response: This has been corrected throughout the manuscript.

Reviewer’s comment: Line 15. “Studies of regulatory guidelines”? What type of studies do you mean? This is better explained in the Intro, but somehow make it clear which studies you mean. Thank you.

Authors’ response: We refer to the study by Larsen et al., 2020, which describes the existence of studies following OECD TG 422, 416, and 415. This is now clarified and included in the abstract.

Reviewer’s comment:  Line 38. Simply define the technology. The statement is inflated.

Authors’ response: The first sentence has been deflated and simplified and now reads: “Nanotechnology promises scientific advances in many fields”. See line 36.

Reviewer’s comment: First paragraph. Delete the “etc.”

Authors’ response: Done.

Reviewer’s comment: Line 56. However, the safety of an application should be warranted before a used is approved, including in occupational, environmental, or consumer uses.

Authors’ response: Done.

Reviewer’s comment:  Line 58. Harmonize.

Authors’ response: Done.

Reviewer’s comment: Line 60. Define OECD-TG.

Authors’ response: The OECD-TG are 415, 416, 421, 422 and 443. This is now indicated in line 59

Reviewer’s comment: Line 64-66. Clarify the writing.

Authors’ response: This sentence has been rewritten to read as follows: “It is worth noting that new approaches based on cellular test methods may also be relevant to support hazard identification of NPs and to reduce animal testing specifically for fertility and reproductive toxicity, as these OECD TGs require the largest number of animals of those used to test for human hazards”. See lines 61-64.

Reviewer’s comment: Around line 105. Could you include more basic information as context for the glia? Just a few sentences even more basic for the readers. For instance, on the internet “Glia, also called glial cells or neuroglia, are non-neuronal cells in the central nervous system and the peripheral nervous system that do not produce electrical impulses. The neuroglia make up more than one half the volume of neural tissue in our body. They maintain homeostasis, form myelin in the peripheral nervous system, and provide support and protection for neurons. In the central nervous system, glial cells include oligodendrocytes, astrocytes, ependymal cells and microglia, and in the peripheral nervous system they include Schwann cells and satellite cells.” Thus, I am suggesting a more basic topic sentence about this interesting cell type.

Authors’ response: The following text has been introduced in line 104: “Glia are non-electrical cells found in the nervous system and consist of several cell types such as astrocytes (involved in modulating synaptic transmission), microglia (specialized macrophages that protect central nervous system neurons), and oligodendrocytes (involved in myelination) [12]. Glia represents approximately 80% of the cells in the human brain and are involved in the homeostasis of the vast majority of neurobiological processes [13]”.

Reviewer’s comment: Line 112. Other evidence supporting the hypothesis that glia regulate (glia is plural) GnRH neurons is that selective….mice caused…

Authors’ response: Done.

Reviewer’s comment: Line 160. “cell viability by 10%”

Authors’ response: Done.

Reviewer’s comment: Table 2: Define GO:xxx. Biological process numbers? Database should be included. Be sure to cite Panther database and url as needed throughout manuscript.

Authors’ response: GO is by Gene Ontology and XXX is the accession number. The GO abbreviation is already defined the first time that the term is used (line 177) and before Table 2.  All cites GO XXX are hyperlinked to the database for more accessibility of the readers to the specific biological term. We have introduced PANTHER database as reference and is now linked in the reference list (reference 20).

Reviewer’s comment: Fig. 3. Describe that the numbers in the Venn diagram mean.

Authors’ response: Done. The figure legend states: “Venn diagram of DEGs in the T98G cells exposed to TiO2-, Cd/Se-QDs and Ag-NPs”.

Reviewer’s comment: Line 286. ..did not realize the potential importance of this finding.

Authors’ answer: Done.

Reviewer’s comment: One additional thing the authors might consider is to list, say in Table 6, some of the main uses (and cite) for these NPs.

Authors’ answer: We respectfully disagree with the reviewer on the convenience of expanding the manuscript with an additional table. However, we agree with him/her that it would be appropriate for the reader to know the uses of the nanoparticles analyzed in the text. Therefore, we have added these uses in lines 126-131.

Reviewer’s comment: Very interesting work

Authors’ answer: Thank you very much.

Reviewer 2 Report

Methods could be written in more detail, but now it's acceptable.

Author Response

Reviewer’s comment: Methods could be written in more detail, but now it's acceptable.

Authors’ response: Thank you very much for your comment. We have provided more detailed information about the methods. Please, see lines 389-402.